# BiGain: Unified Token Compression for Joint Generation and Classification

## Abstract

Acceleration methods for diffusion models (e.g., token merging or downsampling) typically optimize for synthesis quality under reduced compute, yet they often ignore the model's latent discriminative capacity. We revisit token compression with a joint objective and present **BiGain**, a training-free, plug-and-play framework that preserves generation quality while markedly improving classification in accelerated diffusion models. Our key insight is frequency separation: mapping feature-space signals into a frequency-aware representation disentangles fine detail from global semantics, enabling compression that respects both generative fidelity and discriminative utility. BiGain reflects this principle with two frequency-aware operators: (1) *Laplacian-gated token merging*, which encourages merges among spectrally smooth tokens while discouraging merges of high-contrast tokens, thereby retaining edges and textures; and (2) *Interpolate–Extrapolate KV Downsampling*, which downsamples keys/values via a controllable interextrapolation between nearest and average pooling while keeping queries intact, thereby conserving attention precision without retraining. Across DiT- and U-Net–based backbones and multiple datasets of ImageNet-1K, ImageNet-100, Oxford-IIIT Pets, and COCO-2017, our proposed operators consistently improve the speed–accuracy trade-off for diffusion-based classification, while maintaining, sometimes even enhancing generation quality under comparable acceleration. For instance, on ImageNet-1K, with a token merging ratio of 70% on Stable Diffusion 2.0, BiGain improves classification accuracy by **7.1%** while also reduces FID for generation by 0.56 (**3.1%**). Our comprehensive analyses indicate that balanced spectral retention, preserving high-frequency detail alongside low/mid-frequency semantic content is a reliable design rule for token compression in diffusion models. To our knowledge, BiGain is the first framework to jointly study and advance both generation and classification under accelerated diffusion, offering a practical way to deployable, dual-purpose generative systems.

## 1 Introduction

Diffusion models (Ho et al., 2020; Song et al., 2020; Rombach et al., 2022) have become the backbone of modern generative systems, yet their computational footprint during sampling has motivated a surge of acceleration techniques such as token merging (Bolya et al., 2023) and spatial downsampling (Smith et al., 2024). Nearly all of these methods are evaluated (and often tuned) primarily for generation/synthesis fidelity under reduced compute (e.g., keeping FID or perceptual quality stable while cutting FLOPs). This single-objective perspective overlooks an increasingly important use case: the same diffusion backbones are potentially and routinely repurposed for downstream recognition, either through linear probes on intermediate features, feature distillation into smaller classifiers (Tang et al., 2023; Meng et al., 2024), or diffusion-based classification protocols (Li et al., 2023; Clark & Jaini, 2023). In practice, we observe that accelerations that "barely hurt" generation can dramatically weaken discriminative performance.

We argue that token compression should be rethought as a joint optimization problem that simultaneously safeguards generative fidelity and discriminative utility. Empirically, naive compression tends to remove precisely those structures that recognition benefits from (edge/texture cues, small objects, high-contrast boundaries), even when global appearance, and thus visual content remains complete. This creates a gap between what "looks good" and what "classifies well". To bridge this

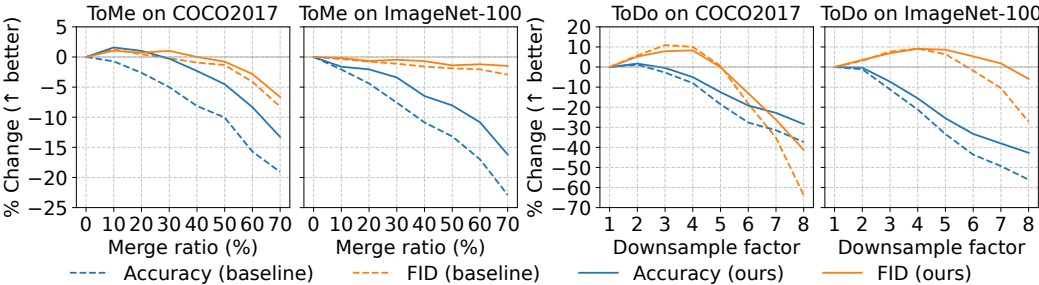

Figure 1: Impact of token compression on diffusion models as our motivation on COCO2017 and ImageNet-100. **Left:** ToMe (Bolya et al., 2023) (baseline) vs. Laplacian-Gated Merge (ours) as the merge ratio increases. **Right:** ToDo (Smith et al., 2024) (baseline) vs. Interpolate–Extrapolate KV-Downsampling (ours) as the downsample factor grows. Curves report percent change relative to the uncompressed model (↑ better; for FID we plot FID%). **Blue:** classification accuracy. **Orange:** generation quality (FID). Baseline compression degrades classification much earlier and faster than generation, sometimes collapsing at extreme sparsity (e.g., COCO2017), whereas our frequency-aware operators consistently curb the classification drop while keeping generation competitive. All experiments in this figure use Stable Diffusion 2.0.

gap, we seek a compression principle that respects the complementary spectral needs of the two capabilities instead of privileging only synthesis. As shown in Fig. 1, baseline compression harms classification accuracy earlier and more sharply than synthesis, sometimes collapsing at extreme sparsity (e.g., COCO2017), whereas our approach consistently mitigates the accuracy drop while keeping generation competitive.

In diffusion classifiers, early denoising emphasizes low frequencies and late steps emphasize high frequencies, and predictions are aggregated across timesteps. Thus token compression must keep both bands and be temporally consistent to avoid excess Monte-Carlo variance. We therefore use heuristics that jointly retain high/low frequencies and apply consistent compression schedules across timesteps. To reflect this, our key insight is frequency separation. Mapping signals of intermediate features into a frequency-aware representation disentangles high-frequency detail (edges, fine textures) from low–mid frequency content (shapes, layouts, semantics). This view yields a simple design rule: balanced spectral retention to preserve the high-frequency components that anchor recognition while maintaining the low–mid bands that support coherent generation. Guided by this principle, compression can prune redundancy without disproportionately harming either side.

In this work, we propose **BiGain**, a training-free, plug-and-play framework composed of two operators. The first, *Laplacian-gated token merging*, computes local Laplacian magnitudes to guide merging: it encourages merges among spectrally smooth tokens and discourages merges of detail-carrying high-contrast tokens. This helps to retain edges and textured micro-structures that classifiers rely on, yet still collapses redundant flat regions to save compute. Crucially, the operator is architecture-agnostic and can be inserted at inference time without retraining. Second, *Interpolate–Extrapolate KV-downsampling* targets attention compute by downsampling keys/values with a controllable interpolation/extrapolation between nearest-neighbor and average pooling (IE-KVD), while leaving queries intact. Keeping queries at full resolution preserves the model's ability to localize and attend precisely, whereas the KV shrinkage reduces memory and FLOPs smoothly, allowing a tunable speed–accuracy trade-off. The two operators are complementary: Laplacian gating biases compression away from detail tokens, and KV downsampling preserves attentional precision, together enabling compression that respects both tasks.

Across DiT- and U-Net–based backbones and multiple datasets, BiGain consistently improves the speed–accuracy trade-off for diffusion-based classification while maintaining generation quality under comparable acceleration, often matching or slightly surpassing the synthesis fidelity of prior accelerations that do not consider recognition at all. Ablations confirm the necessity of frequency awareness: removing Laplacian gating disproportionately hurts classification, and downsampling KV in the frequency domain is necessary for generation. These results suggest that respecting a balanced spectrum is a robust guiding principle for token compression.

Our contributions of this work are:

- **BiGain** reframes token compression for diffusion models as a bi-objective problem and offers a practical, training-free solution.

- To our knowledge, it is the first framework to jointly study and advance both generation and classification under acceleration of generative models.

- Beyond throughput and recognition gains, our study provides practical design guidance in a frequency-aware regime, merges where signals are smooth, downsamples $KV$ while preserving $Q$ that informs future compression for deployable, dual-purpose generative models.

## 2 RELATED WORK

**Acceleration of Diffusion Models.** The iterative nature of diffusion models has spurred methods that reduce the *number of steps* rather than alter the backbone. DDIM (Song et al., 2020) introduces non-Markovian sampling to take larger steps, and high-order solvers such as DPM-Solver (Lu et al., 2022) further shrink function evaluations while preserving fidelity. Progressive Distillation (Salimans & Ho, 2022) compresses a teacher into a student that matches quality with fewer steps. These techniques largely treat the denoiser architecture as fixed and are thus orthogonal to our approach, which targets *intra-step* compute via token compression. Meanwhile, pruning for diffusion (Zhu et al., 2024; Castells et al., 2024; Fang et al., 2023) has also been explored. For example, Diff-Pruning (Fang et al., 2023) uses a Taylor expansion over pruned timesteps, discarding non-contributory steps and aggregating informative gradients to rank important weights. DiP-GO (Zhu et al., 2024) casts pruning as subnet search: it builds a SuperNet with backup connections over similar features and trains a plug-in pruner with tailored losses to identify redundant computation.

**Token Reduction for Diffusion.** Token reduction addresses the quadratic cost of attention by removing or merging redundant tokens. TokenLearner (Ryoo et al., 2021) learns a small set of summary tokens, while training-free strategies like ToMe (Bolya et al., 2023) greedily merge similar tokens with minimal accuracy loss. Recent works adapt these ideas to diffusion backbones: ToMeSD (Bolya et al., 2023) merges U-Net tokens at inference to accelerate Stable Diffusion, and complementary efforts explore structured pruning/sparsity for Diffusion Transformers (Peebles & Xie, 2023). Prior art primarily optimizes *generation* speed–quality trade-offs and typically evaluates synthesis metrics; our method is also training-free and drop-in, but is explicitly designed to preserve generative fidelity *and* discriminative utility through frequency-aware compression.

**Diffusion as a Discriminative Learner, and the Open Gap.** Beyond synthesis, diffusion models provide strong features for recognition (Li et al., 2023; Clark & Jaini, 2023). Diffusion-classifier frameworks use a pre-trained denoiser for per-class scoring or for feature extraction with a lightweight head, yielding competitive image classification (Russakovsky et al., 2015; Chen et al., 2023). However, the interaction between *token reduction* and *discriminative performance* has been largely overlooked: accelerations that barely hurt synthesis can severely degrade classification. Our work sits at this intersection. We study how token compression affects both capabilities across U-Net/DiT backbones and introduce a frequency-aware, training-free framework that maintains generation quality while markedly improving diffusion-based classification.

## 3 METHODOLOGY

We first revisit token reduction for diffusion models from a *bi-objective* viewpoint: preserve generative fidelity *and* discriminative utility. After reviewing the denoising diffusion setup and the diffusion-classifier decision rule, we formalize shape-preserving token reduction and introduce two training-free, plug-in operators that are *frequency-aware*: (i) **Laplacian-gated token merging** (L-GTM) and (ii) **Interpolate–Extrapolate KV-downsampling** (IE-KVD). Both operators avoid cross-timestep caching, which is incompatible with diffusion classification, and can be scheduled across timesteps/layers without retraining.

### 3.1 PRELIMINARIES

A diffusion model (Ho et al., 2020; Song et al., 2020) specifies the forward (noising) process

$$q(\mathbf{x}_t \mid \boldsymbol{x}_0) = \mathcal{N}\big(\mathbf{x}_t;\ \sqrt{\bar{\alpha}_t}\,\boldsymbol{x}_0,\ (1-\bar{\alpha}_t)\,\boldsymbol{I}\big), \quad \mathbf{x}_t = \sqrt{\bar{\alpha}_t}\,\boldsymbol{x}_0 + \sqrt{1-\bar{\alpha}_t}\,\epsilon,\ \ \epsilon \sim \mathcal{N}(\mathbf{0}, \boldsymbol{I}), \quad (1)$$

where $\boldsymbol{x}_0$ is the real clean data, $\mathbf{x}_t$ is its noisy version at step $t$, and $\epsilon$ is standard Gaussian noise. The scalar $\bar{\alpha}_t = \prod_{i=1}^{t} \alpha_i$ defines the cumulative noise schedule: a smaller $\bar{\alpha}_t$ means heavier corruption. Thus, each $\mathbf{x}_t$ is a linear combination of the original signal $\boldsymbol{x}_0$ and the noise $\epsilon$.

The denoiser $\epsilon_{\boldsymbol{\theta}}$ is trained in the noise-prediction parameterization,

$$\epsilon_{\boldsymbol{\theta}}(\mathbf{x}_t, c, t) \approx \epsilon, \quad \mathcal{L}(\boldsymbol{\theta}) = \mathbb{E}[\|\epsilon - \epsilon_{\boldsymbol{\theta}}(\mathbf{x}_t, c, t)\|_2^2], \tag{2}$$

where $c$ denotes an optional conditioning variable (e.g., class label or text prompt).

The network $\epsilon_{\boldsymbol{\theta}}$ learns to recover the exact Gaussian noise injected in the forward process. This training objective is equivalent to maximizing a variational lower bound (ELBO) on the data likelihood. It provides two core capabilities: (i) *iterative generative sampling* by reversing the noising process, and (ii) *per-class scoring for classification* by checking which conditioning $c$ yields the lowest prediction error.

### 3.1.1 DIFFUSION CLASSIFIER

**Decision rule.** Given $\boldsymbol{x}$ and class set $\mathcal{C}$, draw a *shared* Monte Carlo set $\mathcal{S}_{\mathrm{MC}} = \{(t_s, \epsilon_s)\}_{s=1}^{S}$ for all classes. Define:

$$\mathbf{x}_{t_s} = \sqrt{\bar{\alpha}_{t_s}}\,\boldsymbol{x} + \sqrt{1 - \bar{\alpha}_{t_s}}\,\epsilon_s, \qquad \ell(\boldsymbol{x}, c; t_s, \epsilon_s) = \left\|\epsilon_s - \epsilon_{\boldsymbol{\theta}}(\mathbf{x}_{t_s}, c, t_s)\right\|_2^2. \tag{3}$$

Here $\bar{\alpha}_{t_s}$ and $\epsilon_s$ are as defined in the diffusion setup above, and $\epsilon_{\boldsymbol{\theta}}$ is the same denoiser evaluated under class conditioning $c$. Thus $\ell(\boldsymbol{x}, c; t_s, \epsilon_s)$ quantifies how well conditioning on $c$ explains the corruption realized at $(t_s, \epsilon_s)$. The class score and prediction are:

$$\widehat{L}(\boldsymbol{x}, c) = \tfrac{1}{S}\sum_{s=1}^{S} \ell(\boldsymbol{x}, c; t_s, \epsilon_s), \qquad \hat{y}(\boldsymbol{x}) = \arg\min_{c \in \mathcal{C}} \widehat{L}(\boldsymbol{x}, c). \tag{4}$$

Sharing $(t_s, \epsilon_s)$ across classes yields a paired-difference estimate of the ELBO for $\log p_{\boldsymbol{\theta}}(\boldsymbol{x} \mid c)$ without changing the decision rule.

**Adaptive evaluation (staged pruning).** For large $|\mathcal{C}|$, uniform evaluation is costly. We therefore allocate computation in $N_{\mathrm{stages}}$ rounds with cumulative budgets $(T_1, \ldots, T_{N_{\mathrm{stages}}})$ and keep-counts $(K_1, \ldots, K_{N_{\mathrm{stages}}})$ (also see Appendix C): at stage $i$, each surviving class accrues evaluations up to $T_i$, then only the $K_i$ lowest-score classes are retained for the next stage. This staged pruning discards unlikely classes early and concentrates samples on plausible ones, reducing wall-clock compute while leaving the final decision $\arg\min_c \widehat{L}(\boldsymbol{x}, c)$ unchanged.

### 3.1.2 ATTENTION AND SHAPE-PRESERVING TOKEN REDUCTION

Let the denoiser operate on $N$ latent tokens $\boldsymbol{X} \in \mathbb{R}^{N \times d}$ (rows $\boldsymbol{x}_i$). A standard self-attention block forms:

$$\boldsymbol{Q} = \boldsymbol{X}\boldsymbol{W}_Q, \qquad \boldsymbol{K} = \boldsymbol{X}\boldsymbol{W}_K, \qquad \boldsymbol{V} = \boldsymbol{X}\boldsymbol{W}_V, \qquad \mathrm{Attn}(\boldsymbol{X}) = \mathrm{softmax}\left(\tfrac{\boldsymbol{Q}\boldsymbol{K}^{\top}}{\sqrt{d_k}}\right)\boldsymbol{V}. \tag{5}$$

To accelerate while keeping the output length $N$, we use a shape-preserving reduction $\boldsymbol{M} \in \mathbb{R}^{N' \times N}$ with $N' < N$, and, if queries are reduced, an unmerge operator $\boldsymbol{U} \in \mathbb{R}^{N \times N'}$:

$$\boldsymbol{X} \xrightarrow{\tilde{\boldsymbol{X}} = \boldsymbol{M}\boldsymbol{X}} \boldsymbol{Z} = F(\tilde{\boldsymbol{X}}) \xrightarrow{\bar{\boldsymbol{X}} = \boldsymbol{U}\boldsymbol{Z}} \bar{\boldsymbol{X}} \in \mathbb{R}^{N \times d}. \tag{6}$$

We consider two concrete, training-free instances below.

### 3.2 BIGAIN: FREQUENCY-AWARE TOKEN COMPRESSION

Our central design rule is **balanced spectral retention**: preserve high-frequency detail (edges/textures) and low/mid-frequency content (global semantics). We instantiate this via two complementary operators.

### 3.2.1 LAPLACIAN-GATED TOKEN MERGING (L-GTM)

**Goal.** Merge spectrally smooth tokens while discouraging merges of high-contrast tokens.

**Spectral proxy.** Reshape $\boldsymbol{X} \in \mathbb{R}^{N \times d}$ to $\boldsymbol{X} \in \mathbb{R}^{H \times W \times C}$ ($C = d$) and compute a per-location frequency magnitude via a spatial Laplacian:

$$\boldsymbol{F} = \text{Reduce}_c(|\boldsymbol{X} * \boldsymbol{L}|), \quad \boldsymbol{L} = \begin{bmatrix} 0 & 1 & 0 \\ 1 & -4 & 1 \\ 0 & 1 & 0 \end{bmatrix}, \quad \boldsymbol{F} \in \mathbb{R}^{H \times W}. \tag{7}$$

Here $\text{Reduce}_c$ is channel-wise aggregation (e.g., mean or $\ell_2$). $\boldsymbol{L}$ denotes the *Laplacian kernel*, a finite approximation of the second-order derivatives of features in the spatial dimensions (height and width). It is used to measure the degree of difference with respect to the local neighborhood.

**Gated merging.** Let $s_{ij} = \boldsymbol{F}_{ij}$. In each grid, we take the tokens with the lowest $s_{ij}$ values as the destination set $\mathcal{A}$ (low-frequency anchors), and all remaining tokens as the source set $\mathcal{B}$. We then merge the top $r\%$ most similar source–destination pairs by equal-weight averaging. The resulting anchors form the reduced sequence $\tilde{\boldsymbol{X}}$, which defines the merge operator $\boldsymbol{M}$; if needed, $\boldsymbol{U}$ restores shape by broadcasting averaged values back to removed indices. This encourages compression among spectrally smooth tokens while leaving high-frequency tokens largely intact.

**Compute.** When $\boldsymbol{M}$ reduces $\boldsymbol{Q}, \boldsymbol{K}, \boldsymbol{V}$ to $N$' tokens, attention costs shrink from $\mathcal{O}(N^2 d)$ to $\mathcal{O}(N'^2 d)$. L-GTM is architecture-agnostic and training-free, we never touch class tokens in DiT nor time/text conditioning tokens in U-Net cross-attention.

**Blockwise ABM (Adaptive Block Merging) — a fast variant.** For additional efficiency, we introduce a tiled variant that pools an $s \times s$ block $t$ only if $\phi(t) = \max_{(i,j) \in t} \boldsymbol{F}_{ij} < \tau$ (with $\tau$ as a quantile of $\boldsymbol{F}$). Pooled tokens are averaged, others are kept verbatim. ABM is a drop-in replacement for L-GTM in high-resolution stages.

### 3.2.2 INTERPOLATE–EXTRAPOLATE KV-DOWNSAMPLING (IE-KVD)

**Goal.** Reduce attention cost by downsampling keys/values while keeping queries intact to preserve localization and alignment.

**Operator.** Given a stride $s$ and reduced grid $\tilde{H} \times \tilde{W}(\tilde{N} = \tilde{H}\tilde{W} \ll N)$, define a per-site interpolator/extrapolator between nearest and average pooling:

$$\mathcal{D}_{\alpha,s}(\boldsymbol{Z})[i] = \alpha \, \boldsymbol{Z}[\text{nearest}(i)] + (1-\alpha) \frac{1}{|\mathcal{N}_s(i)|} \sum j \in \mathcal{N}s(i)\boldsymbol{Z}[j], \quad \alpha \in \mathbb{R}. \tag{8}$$

We set $\tilde{\boldsymbol{K}} = \mathcal{D}_{\alpha,s}(\boldsymbol{K})$ and $\tilde{\boldsymbol{V}} = \mathcal{D}_{\alpha,s}(\boldsymbol{V})$, while $\boldsymbol{Q}$ remains full-resolution. The attention then costs $\mathcal{O}(N\tilde{N}d)$ and preserves output length $N$.

Preserving $\boldsymbol{Q}$ maintains fine-grained receptive fields for every output token, which stabilizes synthesis, and critically retains discriminative cues (edge/texture) in diffusion classification, where per-token attention precision matters for the MC scoring rule.

### 3.3 COMPATIBILITY WITH DIFFUSION CLASSIFICATION

Our operators are *timestep-local*, deterministic given $\boldsymbol{X}$, and do not rely on cross-timestep caches. They therefore integrate seamlessly with the diffusion-classifier decision rule in Sec. 3.1.1: all classes receive identical $(t_s, \epsilon_s)$ and identical compression schedules, so the paired-difference estimator remains valid. In practice, we reduce per-class FLOPs *and* improve accuracy relative to baselines that focus solely on synthesis quality.

## 4 EXPERIMENTS

### 4.1 EXPERIMENTAL SETUP

**Models.** We test our **BiGain** on two representative diffusion models: **Stable Diffusion v2.0** (Rombach et al., 2022) (UNet-based latent diffusion with text conditioning) and **DiT-XL/2** (Peebles &

Xie, 2022) (Transformer backbone), using official pretrained weights. Diffusion classifiers require a noise predictor $\hat{\epsilon}_\theta(x_t, t)$.

**Datasets and Metrics.** For classification we use comprehensive datasets of `ImageNet-1K` (Russakovsky et al., 2015), `ImageNet-100` (Tian et al., 2020), `Oxford-IIIT Pets` (Parkhi et al., 2012), and `COCO2017` (Lin et al., 2014). Following Li et al. (2023), we evaluate on a 2,000-image validation subset for ImageNet-1K (linear cost in $|\mathcal{C}|$); full validation splits are used elsewhere. We report Top-1 accuracy for single-label datasets and Top-1 precision plus mAP (macro) for multi-label COCO. For generation we evaluate on COCO2017 captions, ImageNet-100, and ImageNet-1K, reporting FID metric. DiT-XL/2 is evaluated only on ImageNet datasets (class-index conditioning, no free-form prompts), while Stable Diffusion v2.0 is evaluated on all datasets using text class prompts. Efficiency is reported as sparsity and FLOPs (both total and attention FLOPs).

**Implementation Details.** Considering the unified timestep policy, also to make generation and diffusion-classifier settings directly comparable, we apply the same token-reduction policy at every denoising step $t$. We do not cache merge pairings or pooling indices across timesteps; all reductions are recomputed per step and per block. This avoids $t$-dependent artifacts for synthesis and, because the diffusion classifier is a Monte-Carlo estimator over $(t, \epsilon)$, keeps the schedule temporally consistent, reducing unnecessary variance.

## 4.2 COMPARISONS TO THE STATE-OF-THE-ART APPROACHES

Table 1 presents the comparisons with state-of-the-art approaches on Oxford-IIIT Pets using Top-1 accuracy under ~10% FLOPs reduction. The no-acceleration baseline is 81.03%. Token-merging/pruning baselines suffer large drops: ToMe (8.07%) and SiTo (12.19%), with pruning methods DiP-GO (4.50%) and MosaicDiff (3.65%), showing that compression tuned for synthesis often harms recognition. Our Laplacian-gated merging (**BiGain$_{TM}$**) retains far more accuracy (78.38%, 2.65% drop), cutting the loss by 40~80% vs. these methods at matched FLOPs. In the downsampling regime (14.2% FLOPs), ToDo slightly decreases the accuracy (-1.88%), while our Interpolate–Extrapolate KV-downsampling (**BiGain$_{TD}$**) is the best overall (79.90%, only 1.13% drop), also with much better generation ability than ToDo, as we will discuss later. Overall, **BiGain** delivers the strongest classification under comparable compute.

Table 1: Classification accuracy (Acc@1) on Pets dataset under similar FLOPs reduction.

| Method | Acceleration Type | FLOPs Reduction ↑ | Acc@1 ↑ (%) | Δ vs. Baseline ↓ |
|---|---|---|---|---|
| Baseline (No Accel.) | None | – | 81.03 | – |
| ToMe (Bolya et al., 2023) | Token Merging/Pruning | 10% | 72.96 | ↓ 8.07 |
| DiP-GO (Zhu et al., 2024) | Model Pruning | 10% | 76.53 | ↓ 4.50 |
| SiTo (Zhang et al., 2025) | Token Merging/Pruning | 7% | 68.84 | ↓ 12.19 |
| MosaicDiff (Guo et al., 2025) | Model Pruning | 10% | 77.38 | ↓ 3.65 |
| **BiGain$_{TM}$ (Ours)** | Token Merging/Pruning | 10% | **78.38** | ↓ **2.65** |
| ToDo (Smith et al., 2024) | Token Downsampling | 14.2% | 79.15 | ↓ 1.88 |
| **BiGain$_{TD}$ (Ours)** | Token Downsampling | 14.2% | **79.90** | ↓ **1.13** |

## 4.3 CLASSIFICATION VS. GENERATION EXPERIMENTS

We report classification and generation comparisons under Token Downsampling in Table 2 (SD-2.0 backbone) and Table 3 (DiT-XL/2). As shown, our method consistently outperforms the baseline, and the advantage becomes more pronounced as the downsampling ratio increases. The same trend holds for generation: with higher downsampling factors, our approach yields increasingly better results. We further observe (Table 3) that the ToDo method performs very poorly on the DiT-XL/2 model, whereas our method remains stable and effective on this backbone. Furthermore, with relatively small downsampling ($2\times$), our method surpasses the original unaccelerated model in both classification and generation.

For Token Merging, classification and generation comparisons are reported in Table 4 (SD-2.0 backbone) and Table 5 (DiT-XL/2). The experimental results mirror those under downsampling: as the merging ratio increases (i.e., with more aggressive pruning), our method achieves substantially better performance than the baseline. In particular, our classification accuracy significantly surpasses ToMe, while our generation capability also exceeds it. These results highlight the dual advantages of our approach in both classification and generation.

Table 2: SD-2.0 **Token Downsampling**: Classification (Acc@1 on Pets, ImageNet-100/1K; Acc@1 and mAP on COCO-2017) and generation fidelity (FID ↓) vs. downsampling factor. For classification, we fix the interextrapolation factor at 0.9 across all timesteps to ensure stability. For generation, we linearly vary the factor from 0.8 (early steps) to 1.2 (later steps), shifting emphasis from low- to high-frequency information. Gray color indicates the same generation results as the above group.

| Method | No Accel. | Classification ↑ (TD×) | | | | | | | No Accel. | Generation ↓ (TD×) | | |
|---|---|---|---|---|---|---|---|---|---|---|---|---|
| | | 2× | 3× | 4× | 5× | 6× | 7× | 8× | | 2× | 3× | 4× |
| **Pets** | | | | | | | | | | | | |
| Avg-pooling (baseline) | | 77.02 | 73.45 | 71.26 | 69.00 | 67.56 | 66.66 | 65.13 | | 38.50 | 39.42 | 39.74 |
| ToDo (Smith et al., 2024) | 81.03 | 81.30 | 79.15 | 77.46 | 72.74 | 66.74 | 62.87 | 56.16 | 35.01 | 33.52 | 32.38 | 31.48 |
| **BiGain$_{TD}$**(Ours) | | **81.52** | **79.91** | **78.03** | **74.92** | **70.86** | **69.33** | **66.03** | | **32.19** | **30.44** | **29.21** |
| Δ↑ | | ↑0.22 | ↑0.76 | ↑0.57 | ↑2.18 | ↑4.12 | ↑6.46 | ↑9.87 | | ↓1.33 | ↓1.94 | ↓2.27 |
| **ImageNet-100** | | | | | | | | | | | | |
| Avg-pooling (baseline) | | 58.50 | 49.52 | 45.54 | 40.96 | 38.74 | 38.12 | 37.40 | | 19.31 | 23.08 | 26.59 |
| ToDo (Smith et al., 2024) | 73.12 | 72.30 | 64.96 | 57.62 | 48.70 | 41.22 | 37.04 | 32.12 | 17.64 | 16.86 | 15.93 | 15.63 |
| **BiGain$_{TD}$**(Ours) | | **72.88** | **67.78** | **61.72** | **54.48** | **48.78** | **45.30** | **41.90** | | **16.46** | **15.46** | **15.46** |
| Δ↑ | | ↑0.58 | ↑2.82 | ↑4.10 | ↑5.78 | ↑7.56 | ↑8.26 | ↑9.78 | | ↓0.40 | ↓0.47 | ↓0.17 |
| **COCO-2017** | | | | | | | | | | | | |
| *Acc@1* Avg-pooling (baseline) | | 62.98 | 55.94 | 52.46 | 48.74 | 46.88 | 47.38 | 46.74 | | 30.52 | 35.92 | 41.23 |
| *Acc@1* ToDo (Smith et al., 2024) | 70.84 | 71.66 | 68.90 | 65.16 | 57.70 | 51.26 | 48.52 | 44.40 | 26.79 | 25.26 | 23.86 | 24.10 |
| *Acc@1* **BiGain$_{TD}$**(Ours) | | **72.04** | **70.52** | **67.28** | **61.98** | **57.26** | **54.66** | **50.72** | | **24.29** | **23.17** | **24.05** |
| Δ↑ | | ↑0.38 | ↑1.62 | ↑2.12 | ↑4.28 | ↑6.00 | ↑6.14 | ↑6.32 | | ↓0.97 | ↓0.69 | ↓0.05 |
| *mAP* Avg-pooling (baseline) | | 44.25 | 40.96 | 38.98 | 36.89 | 35.77 | 35.79 | 35.38 | | 30.52 | 35.92 | 41.23 |
| *mAP* ToDo (Smith et al., 2024) | 46.01 | 46.59 | 45.56 | 44.07 | 40.31 | 36.95 | 35.50 | 33.34 | 26.79 | 25.26 | 23.86 | 24.10 |
| *mAP* **BiGain$_{TD}$**(Ours) | | **46.97** | **46.28** | **44.81** | **42.54** | **40.28** | **38.82** | **36.93** | | **24.29** | **23.17** | **24.05** |
| Δ↑ | | ↑0.38 | ↑0.72 | ↑0.74 | ↑2.23 | ↑3.33 | ↑3.32 | ↑3.59 | | ↓0.97 | ↓0.69 | ↓0.05 |

Table 3: DiT-XL/2 **Token Downsampling**: Classification (Acc@1) and generation fidelity (FID ↓) vs. downsampling factor. For both classification and generation, we fix the interpolate-extrapolate factor at 0.1 across all timesteps. TD Factor: Token Downsampling factor.

| Method | No Accel. | Classification ↑ (TD×) | | | | No Accel. | Generation ↓ (TD×) | | | |
|---|---|---|---|---|---|---|---|---|---|---|
| | | 2× | 3× | 4× | 5× | | 2× | 3× | 4× | 5× |
| **ImageNet-100** | | | | | | | | | | |
| Avg-pooling (baseline) | | 78.34 | 61.04 | 48.40 | 33.26 | | **40.13** | 33.57 | 30.25 | 41.61 |
| ToDo (Smith et al., 2024) | 84.82 | 69.34 | 8.46 | 4.74 | 3.32 | 41.37 | 40.48 | 190.18 | 206.52 | 215.04 |
| **BiGain$_{TD}$ (Ours)** | | **78.42** | **61.58** | **48.72** | **34.00** | | **40.13** | **32.95** | **29.87** | **40.55** |
| Δ↑ | | ↑9.08 | ↑53.12 | ↑43.98 | ↑30.68 | | ↓0.35 | ↓157.23 | ↓176.65 | ↓174.49 |

## 4.4 ABLATION

**Where to reduce.**[1] As shown in Table 6, we compare applying token reduction to *self-attention only* (SA), *self+cross attentions* (SA+CA), and *self+cross+MLP* (SA+CA+MLP). We find that **SA-only** consistently delivers the best quality–efficiency trade-off: it preserves prompt adherence (avoiding CA degradation) and avoids compounding bias through MLP compression. On Pets, SA-only attains the highest accuracy, while SA+MLP reduces prompt fidelity and SA+CA+MLP further harms fine details. *Conclusion:* we adopt **SA-only** reduction as default for all SD 2.0 experiments.

**How to score tokens**. As shown in Table 7, local frequency cues dominate: Laplacian Filter ($\ell_1$) is best at all merge ratios, outperforming global statistics (norms, channel variance),

Table 7: **Ablation over token scoring heuristics for Stable Diffusion 2.0.** Top-1 accuracy (%) on Pets dataset across merge ratios. Local Laplacian signals outperform global or spectral metrics.

| Scoring method | 0.7 | 0.5 | 0.3 |
|---|---|---|---|
| Global mean deviation | 72.96 | 77.84 | 79.91 |
| $\ell_1$-norm | 73.02 | 77.11 | 79.86 |
| $\ell_2$-norm | 72.72 | 77.95 | 79.61 |
| Channel variance | 73.04 | 77.95 | 79.83 |
| Laplacian Filter $\ell_1$ | **74.63** | **78.38** | **80.40** |
| Laplacian Filter $\ell_2$ | 74.24 | 77.81 | 79.80 |
| DFT spectral centroid | 73.75 | 77.92 | 79.10 |
| DFT amplitude | 73.10 | 77.76 | 79.34 |
| Cosine to neighbors | 74.00 | 78.22 | 79.56 |
| Cosine to global mean | 73.32 | 77.84 | 79.83 |

spectral DFT measures, and cosine similarity by 0.3∼1.9%. This supports our frequency-aware design and motivates using a Laplacian proxy for gated merging. Overall, for SD-2.0, token merging in SA with Laplacian scoring provides the strongest quality–efficiency trade-off under our ablation protocol. The detailed mathmatical formulations of these score heuristics can be seen in Appendix C.2.

---

[1]Unless noted otherwise, ablations are conducted on `Oxford-IIIT Pets` with identical sampling schedules, classifier settings (for classification ablations), and reduction ratios as in the main results.

Table 4: SD-2.0 **Token Merging**: Classification (Acc@1 on Pets, ImageNet-100/1K; Acc@1 and mAP on COCO-2017) and generation fidelity (FID ↓) vs. Token Merging Ratio

| Method | No Accel. | Classification ↑ (Token Merging Ratio) | | | | | | | No Accel. | Generation ↓ (Token Merging Ratio) | | | | | | |
|---|---|---|---|---|---|---|---|---|---|---|---|---|---|---|---|---|
| | | 10% | 20% | 30% | 40% | 50% | 60% | 70% | | 10% | 20% | 30% | 40% | 50% | 60% | 70% |
| **Pets** | | | | | | | | | | | | | | | | |
| ToMe | | 80.10 | 79.88 | 78.44 | 76.42 | 72.96 | 69.93 | 65.76 | | 35.05 | 35.30 | 35.71 | 36.26 | 37.00 | 37.63 | 38.35 |
| BiGain$_{TM}$ (Ours) | 81.03 | **81.16** | **81.16** | **80.40** | **80.07** | **78.38** | **76.04** | **74.63** | 35.01 | **35.00** | **35.12** | **35.01** | **35.99** | **36.52** | **36.99** | **37.73** |
| Δ ↑ | | ↑1.06 | ↑1.28 | ↑1.96 | ↑3.65 | ↑5.42 | ↑6.11 | ↑8.87 | | ↓0.05 | ↓0.18 | ↓0.70 | ↓0.27 | ↓0.48 | ↓0.64 | ↓0.62 |
| **ImageNet-100** | | | | | | | | | | | | | | | | |
| ToMe | | 71.60 | 69.90 | 67.58 | 65.18 | 63.48 | 60.70 | 56.38 | | 41.51 | 41.68 | 41.82 | 42.02 | 42.15 | 42.21 | 42.58 |
| BiGain$_{TM}$ (Ours) | 73.12 | **71.94** | **71.62** | **70.64** | **68.38** | **67.24** | **65.20** | **61.28** | 41.37 | **41.43** | **41.64** | **41.55** | **41.65** | **41.93** | **41.86** | **41.98** |
| Δ ↑ | | ↑0.34 | ↑1.72 | ↑3.06 | ↑3.20 | ↑3.76 | ↑4.50 | ↑4.90 | | ↓0.08 | ↓0.04 | ↓0.27 | ↓0.37 | ↓0.22 | ↓0.35 | ↓0.60 |
| **ImageNet-1K** | | | | | | | | | | | | | | | | |
| ToMe | | 55.50 | 54.25 | 52.35 | 50.65 | 47.55 | 43.55 | 37.35 | | 17.57 | 17.66 | 17.74 | 17.74 | 17.83 | 17.97 | 18.42 |
| BiGain$_{TM}$ (Ours) | 57.05 | **57.25** | **56.50** | **55.80** | **54.80** | **52.50** | **49.10** | **44.50** | 17.64 | **17.54** | **17.48** | **17.52** | **17.53** | **17.58** | **17.69** | **18.08** |
| Δ ↑ | | ↑1.75 | ↑2.25 | ↑3.45 | ↑4.15 | ↑4.95 | ↑5.55 | ↑7.15 | | ↓0.03 | ↓0.18 | ↓0.22 | ↓0.21 | ↓0.25 | ↓0.28 | ↓0.34 |
| **COCO-2017** | | | | | | | | | | | | | | | | |
| Acc@1 \| ToMe | | 70.32 | 68.98 | 67.3 | 65.08 | 63.72 | 59.72 | 57.32 | | 26.45 | 26.68 | 26.85 | 27.04 | 27.15 | 27.89 | 29.00 |
| Acc@1 \| BiGain$_{TM}$ | 70.84 | **71.96** | **71.56** | **70.64** | **69.20** | **67.64** | **64.94** | **61.44** | 26.79 | 26.51 | **26.60** | **26.52** | **26.79** | **27.00** | **27.55** | **28.57** |
| Δ ↑ | | ↑1.64 | ↑2.58 | ↑3.34 | ↑4.12 | ↑3.92 | ↑5.22 | ↑4.12 | | ↑0.06 | ↓0.08 | ↓0.33 | ↓0.25 | ↓0.15 | ↓0.34 | ↓0.43 |
| mAP \| ToMe | | 46.04 | 45.35 | 44.50 | 43.43 | 42.82 | 41.01 | 40.07 | | 26.45 | 26.68 | 26.85 | 27.04 | 27.15 | 27.89 | 29.00 |
| mAP \| BiGain$_{TM}$ | 46.01 | **46.38** | **46.21** | **46.05** | **45.50** | **44.94** | **43.98** | **42.44** | 26.79 | 26.51 | **26.60** | **26.52** | **26.79** | **27.00** | **27.55** | **28.57** |
| Δ ↑ | | ↑0.34 | ↑0.86 | ↑1.55 | ↑2.07 | ↑2.12 | ↑2.97 | ↑2.37 | | ↑0.06 | ↓0.08 | ↓0.33 | ↓0.25 | ↓0.15 | ↓0.34 | ↓0.43 |

Table 5: DiT-XL/2 **Token Merging**: Classification (Acc@1 on Pets, ImageNet-100/1K; Acc@1 and mAP on COCO-2017) and generation fidelity (FID ↓) vs. Token Merging Ratio

| Method | No Accel. | Classification ↑ (Token Merging Ratio) | | | | | | | No Accel. | Generation ↓ (Token Merging Ratio) | | | | | | |
|---|---|---|---|---|---|---|---|---|---|---|---|---|---|---|---|---|
| | | 10% | 20% | 30% | 40% | 50% | 60% | 70% | | 10% | 20% | 30% | 40% | 50% | 60% | 70% |
| **ImageNet-100** | | | | | | | | | | | | | | | | |
| ToMe | | 80.86 | 78.02 | 75.3 | 71.38 | 68.24 | 62.06 | 53.88 | | 41.51 | 41.68 | 41.83 | 42.02 | 42.15 | 42.21 | 42.58 |
| BiGain$_{TM}$ | 84.82 | **83.56** | **82.2** | **79.92** | **77.38** | **73.68** | **68.34** | **61.76** | 47.53 | **41.43** | **41.61** | **41.56** | **41.65** | **41.92** | **41.77** | **41.89** |
| Δ ↑ | | ↑2.70 | ↑4.18 | ↑4.62 | ↑6.00 | ↑5.44 | ↑6.28 | ↑7.88 | | ↓0.08 | ↓0.07 | ↓0.27 | ↓0.37 | ↓0.23 | ↓0.44 | ↓0.69 |

Table 6: **Ablation of token-merging locations in Stable Diffusion 2.0 on Pets.** Self-Attention (SA) is always merged; Cross-Attention (CA) and MLP are toggled. Results reported at merge ratios $r \in \{0.7, 0.5, 0.3\}$. The underlined results indicate the best performance across all configurations.

| | SA only | | | SA+CA | | | SA+MLP | | | SA+CA+MLP | | |
|---|---|---|---|---|---|---|---|---|---|---|---|---|
| **Method** | **0.7** | **0.5** | **0.3** | **0.7** | **0.5** | **0.3** | **0.7** | **0.5** | **0.3** | **0.7** | **0.5** | **0.3** |
| ToMe (Bolya et al., 2023) | 65.76 | 72.96 | 78.44 | 61.68 | 68.41 | 74.46 | 51.43 | 58.71 | 66.35 | 50.86 | 59.53 | 66.20 |
| BiGain$_{TM}$ (Ours) | 74.63 | 78.38 | 80.40 | 73.89 | 78.03 | 79.56 | 68.27 | 74.93 | 77.98 | 68.25 | 74.84 | 77.95 |

## 4.5 ANALYSIS

**Further Speedup.** *Our vanilla Laplacian Merge.* Before the Q/K/V projections, we run a 2-D Laplacian filter on the hidden map to score each token by local frequency (contrast w.r.t. its four neighbors). We then partition the feature map into $s_x \times s_y$ cells; within each cell, low-frequency tokens serve as *destinations* and the remaining *source* tokens are greedily assigned by cosine similarity. Because merging acts like a low-pass filter that can destroy high-freq detail, we restrict merging

Table 8: Further speedup on SD-2.0 **Token Merging**: classification performance vs. merge ratio. Acc@1 for single-label datasets; Acc@1 and mAP for multi-label COCO-2017.

| Dataset | Method | GFLOPs | 10% | 20% | 30% | 40% | 50% | 60% | 70% |
|---|---|---|---|---|---|---|---|---|---|
| Pets | Laplacian Gated Merge | 704.99 | 81.16 | 81.16 | 80.4 | 80.07 | 78.38 | 76.04 | 74.63 |
| | Cached Assignment Merge | 698.88 | 80.29 | 79.97 | 79.89 | 79.01 | 78.11 | 75.91 | 74.49 |
| | Adaptive Block Merge | 695.08 | 80.40 | 80.16 | 79.99 | 79.18 | 77.84 | 75.96 | 74.13 |
| ImageNet-100 | Laplacian Gated Merge | 704.99 | 71.94 | 71.62 | 70.64 | 68.38 | 67.24 | 65.20 | 61.28 |
| | Cached Assignment Merge | 698.88 | 71.76 | 71.16 | 70.44 | 69.38 | 67.78 | 64.56 | 61.28 |
| | Adaptive Block Merge | 695.08 | 72.58 | 71.94 | 70.58 | 70.52 | 68.04 | 65.36 | 60.98 |
| ImageNet-1K | Laplacian Gated Merge | 704.99 | 57.25 | 56.50 | 55.80 | 54.80 | 52.50 | 49.10 | 44.50 |
| | Cached Assignment Merge | 698.88 | 56.30 | 56.05 | 56.05 | 53.15 | 52.30 | 47.90 | 44.60 |
| | Adaptive Block Merge | 695.08 | 56.95 | 56.25 | 56.00 | 54.60 | 51.95 | 48.20 | 44.85 |
| COCO-2017 | Acc@1 \| Laplacian Gated Merge | 704.99 | 71.96 | 71.56 | 70.64 | 69.2 | 67.64 | 64.94 | 61.44 |
| | Acc@1 \| Cached Assignment Merge | 698.88 | 71.72 | 71.40 | 70.22 | 70.02 | 67.94 | 64.70 | 60.88 |
| | Acc@1 \| Adaptive Block Merge | 695.08 | 71.76 | 71.44 | 70.28 | 69.62 | 67.26 | 64.70 | 60.56 |
| | mAP \| Laplacian Gated Merge | 704.99 | 46.38 | 46.21 | 46.05 | 45.50 | 44.94 | 43.98 | 42.44 |
| | mAP \| Cached Assignment Merge | 698.88 | 46.30 | 46.32 | 45.94 | 45.96 | 45.19 | 43.93 | 42.41 |
| | mAP \| Adaptive Block Merge | 695.08 | 46.35 | 46.41 | 45.93 | 45.87 | 45.12 | 43.96 | 42.32 |

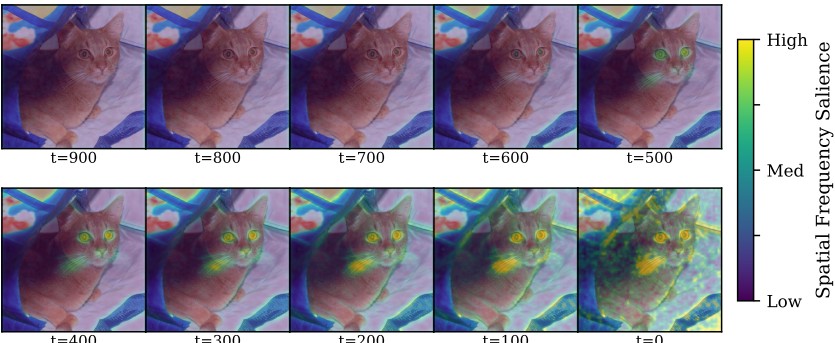

Figure 2: Visualization of our Laplacian-based frequency heuristic on hidden representations from Stable Diffusion-2.0. We probe U-Net at the highest-resolution upsampling stage. The visualization is computed from a noised image without a text prompt, showing the model's intrinsic frequency-aware reconstruction dynamics. To reduce variance, we randomly sample 100 independent noise realizations and visualize the averaged token salience map.

to low-freq tokens only. *Two faster variants.* (1) *Our Cached Assignment Merge*: in the highest-resolution U-Net stages (two Transformer blocks for down sampling and three for up sampling), compute the merge/unmerge map once in the first attention block and reuse it within the stage. (2) *Our Adaptive Block Merge*: after computing Laplacian scores, aggregate them per cell and merge entire low-frequency cells with no per-token matching, yielding extra speed with minimal accuracy loss. As shown in Table 8, both variants closely track Laplacian-Gated Merge across 10∼70% merge ratios across different datasets while providing additional FLOPs savings.

**Visualization.** We compare token-importance maps for generation and classification to reveal their different spectral needs. Overall, as shown in Fig. 2, frequency-aware reduction yields a favorable bias–variance trade-off: retaining low-frequency tokens stabilizes classification, while selectively keeping high-frequency tokens preserves generation quality, making one heuristic effective for both tasks. To illustrate our Laplacian scoring, we probe SD-2.0 at the highest-resolution upsampling block and visualize pre-attention hidden states filtered by a 2-D Laplacian. Maps are averaged over 100 noise draws without a text prompt to reduce variance, to reveal the model's intrinsic frequency sensitivity.

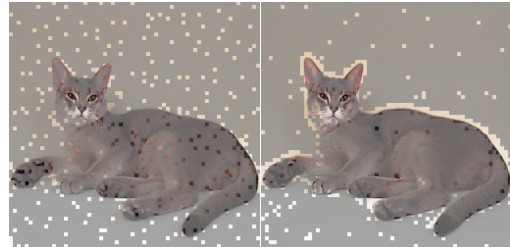

Figure 3: **Comparison of token merging schemes.** Left: ToMe (Bolya et al., 2023); Right: Laplacian-gated token merging (**BiGain**$_{TM}$). Merging is applied with a merge ratio 90% at the highest-resolution latent layer of the U-Net transformer in Stable Diffusion 2.0 at denoising step $t = 200$. Grayscale indicates merged tokens.

In Fig. 3, we compare ToMe *vs.* **BiGain**$_{TM}$ at 90% merge on the highest-resolution U-Net transformer layer at $t =$200 (grayscale = merged). Laplacian-gated merging preserves more class-discriminative structure (e.g., the cat's edges) than standard ToMe.

## 5 CONCLUSION

In this work, we revisited token compression for diffusion models as a bi-objective problem, preserving both generative and discriminative abilities, and introduced **BiGain**, a training-free, cache-free framework built on two frequency-aware operators: *Laplacian-Gated Token Merging* (merge in smooth regions, keep edges) and *Interpolate–Extrapolate KV-Downsampling* (downsample K/V with controllable interextrapolation while keeping Q unchanged). Using DiT/U-Net backbones and multiple datasets, BiGain consistently improves the speed–accuracy trade-off for diffusion-based classification while maintaining, and sometimes even improving generation quality under comparable compute. Our extensive analyses show a simple design rule: balanced spectral retention of high-frequency detail and low/mid-frequency semantics enhances gains. While very aggressive sparsity can still degrade performance, BiGain shifts the Pareto frontier and is deployable as a plug-in.

ETHICS STATEMENT

This work proposes training-free token compression techniques that reduce the compute and energy cost of diffusion models. While efficiency has positive environmental benefits, dual-use risks remain: faster generation and improved classification can be misused for spam or synthetic media. We evaluate only on public benchmarks (ImageNet, COCO, Oxford-IIIT Pets) under their licenses, do not collect personal data, and release no new sensitive datasets. Because compression can subtly shift model outputs, downstream deployments should re-check safety, bias, and content filters, our release will include guidance to toxicity and fairness checks and to respect dataset/model licenses.

REPRODUCIBILITY STATEMENT

BiGain is inference-only and requires no training. We will release code, configs, and scripts to reproduce all tables/figures, including: (i) exact token-reduction schedules per layer; (ii) implementations of Laplacian-Gated Merge and Interpolate–Extrapolate KV-Downsampling; (iii) evaluation code for diffusion classification with fixed seeds, compression settings; (iv) generation metrics (e.g., FID); (v) prompts/class labels used, data preprocessing, and dataset splits; (vi) environment files with library versions and hardware notes; and (vii) FLOPs/sparsity accounting. We use official checkpoints (Stable Diffusion v2.0 and DiT-XL/2) and fix random seeds to ensure run-to-run determinism.

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

# APPENDIX

## CONTENTS

## A    THEORY: FREQUENCY-AWARE TOKEN REDUCTION IMPROVES DIFFUSION CLASSIFICATION VIA VARIANCE CONTROL

**Setting.**    Given an image $x$ and a conditioning $c \in \mathcal{C}$, the Diffusion Classifier (DC) scores each class by the expected $\ell_2$ noise prediction error,

$$S(x, c) \;=\; \mathbb{E}_{t,\epsilon} \left\| \epsilon - \epsilon_\theta(x_t, c) \right\|_2^2, \qquad x_t = \sqrt{\bar{\alpha}_t}\, x + \sqrt{1 - \bar{\alpha}_t}\, \epsilon,$$

and predicts $\hat{c}(x) = \arg\min_{c \in \mathcal{C}} S(x, c)$, using *paired sampling* of $(t, \epsilon)$ across classes. This rule is the uniform-$\ell_2$ ELBO surrogate of (Li et al., 2023) and empirically concentrates accuracy at intermediate timesteps; both the decision rule and the paired-difference rationale.

**Paired difference and a tail bound.**    Fix the true class $c^\star$ and a distractor $\tilde{c}$. For one paired draw $(t, \epsilon)$ define the difference

$$D(t, \epsilon) = \| \epsilon - \epsilon_\theta(x_t, \tilde{c}) \|_2^2 - \| \epsilon - \epsilon_\theta(x_t, c^\star) \|_2^2, \qquad \mu = \mathbb{E}[D], \;\; \sigma^2 = \mathrm{Var}[D].$$

With $N$ paired draws, $\widehat{\Delta}_N = \frac{1}{N} \sum_{i=1}^N D(t_i, \epsilon_i)$ concentrates around $\mu > 0$ for a consistent classifier, and the one-sided Cantelli inequality yields

$$\Pr\left( \widehat{\Delta}_N \leq 0 \right) \;\leq\; \frac{\sigma^2/N}{\mu^2 + \sigma^2/N}.$$

Because the right-hand side is strictly increasing in $r = \sigma/\mu$, improving the bound is equivalent to decreasing $r$.

**A bandwise view.**    Let $\{\phi_k\}$ be an orthonormal 2-D DCT/Fourier basis over the token grid. Expanding the classwise error at $(t, \epsilon)$ produces a band-weighted quadratic form

$$\| \epsilon - \epsilon_\theta(x_t, c) \|_2^2 = \sum_k \omega_k(t) \left| \widehat{\epsilon}(k) - \widehat{\epsilon}_\theta(k; t, c) \right|^2,$$

where $\omega_k(t) \geq 0$ reflects the per-band reliability at step $t$. Writing $D(t, \epsilon) = \sum_k \omega_k(t)\, \Delta_k(t, \epsilon)$ and $w_k = \mathbb{E}_t[\omega_k(t)]$,

$$\mu = \sum_k w_k\, \mu_k, \qquad \sigma^2 = \sum_k w_k^2\, \sigma_k^2 + 2\sum_{i<j} w_i w_j\, \mathrm{Cov}(\Delta_i, \Delta_j).$$

We will use the standard weak-correlation approximation $\sigma^2 \approx \sum_k w_k^2 \sigma_k^2$, which empirically matches DC's paired-difference stability. Low-frequency bands dominate the mean margin $\mu$, whereas high-frequency bands often dominate the variance $\sigma^2$.

**Local token reduction as a spectral operator.**    Consider the attention layer and a window $W$ of tokens. Let $z_i = s_i + n_i$ with structured content $s_i$ and zero-mean perturbation $n_i$. A shape-preserving reduction $P$ maps $\{z_i\}_{i \in W}$ to a representative and, under a local linearization of the block, acts as a *windowed frequency response* $H_P(k)$:

$$\mu' = \sum_k w_k\, H_P(k)\, \mu_k, \qquad \sigma'^2 \approx \sum_k w_k^2\, H_P(k)^2\, \sigma_k^2.$$

Average-type reductions behave as local low-pass filters (attenuate large $k$), while nearest/selection-type reductions preserve amplitude across bands (and can alias under decimation). This model captures precisely how $P$ reshapes the band-weighted paired statistic.

**Main result.**    Let $r = \sigma/\mu$ and $r' = \sigma'/\mu'$, and write the mean and variance changes as

$$\Delta\mu := \mu - \mu' = \sum_k w_k \big( 1 - H_P(k) \big) \mu_k, \qquad \Delta\sigma^2 := \sigma^2 - \sigma'^2 = \sum_k w_k^2 \big( 1 - H_P(k)^2 \big) \sigma_k^2.$$

Since the Cantelli bound is monotone in $r$, improvement is equivalent to $r' < r$. A short calculation gives an exact criterion.

**Theorem 1 (Spectral margin–variance improvement)** *The DC tail bound improves after applying $P$ if and only if*

$$\Delta\sigma^2 \; > \; 2\frac{\sigma^2}{\mu}\,\Delta\mu \; - \; \frac{\sigma^2}{\mu^2}\,(\Delta\mu)^2.$$

*When $\Delta\mu/\mu$ is small, the first-order sufficient condition*

$$\Delta\sigma^2 \; > \; 2\frac{\sigma^2}{\mu}\,\Delta\mu$$

*guarantees $r' < r$.*

*Proof.* The inequality $r'^2 < r^2$ is equivalent to $\frac{\sigma^2-\Delta\sigma^2}{(\mu-\Delta\mu)^2} < \frac{\sigma^2}{\mu^2}$. Clearing denominators and rearranging yields $\Delta\sigma^2\mu^2 - 2\sigma^2\mu\,\Delta\mu + \sigma^2(\Delta\mu)^2 > 0$, which after dividing by $\mu^2$ gives the stated condition; a first-order expansion in $\Delta\mu/\mu$ gives the sufficient bound. □

**Interpretation (spectral balancing).** The improvement condition in Theorem 1 depends on two band-aggregated quantities: the variance shaved off, $\Delta\sigma^2 = \sum_k w_k^2(1 - H_P(k)^2)\sigma_k^2$, and the margin lost, $\Delta\mu = \sum_k w_k(1 - H_P(k))\mu_k$. Because different bands contribute differently to these two terms, achieving $r' < r$ requires a *frequency-selective* (i.e., balanced) response $H_P$: attenuate bands that disproportionately inflate variance while preserving bands that contribute margin—*irrespective of whether those bands are nominally "low," "mid," or "high" frequency*. Practically, this means designing $H_P(k)$ to behave like a bandwise shrinkage rule, with $H_P(k) \approx 1$ where $\mu_k$ dominates $\sigma_k$ (margin-rich bands) and $H_P(k) < 1$ where $\sigma_k^2$ dominates (variance-heavy bands). Such spectral balancing tightens the Cantelli bound by reducing variance without proportionally erasing discriminative content, and it subsumes low-pass filtering as a special case rather than a requirement.

**Assumptions and scope.** The derivation relies on a local linearization at the reduced layer and on a weak cross-band correlation approximation in the paired statistic; both are standard in analyzing attention-layer perturbations and match the behavior we observe when DC is implemented with paired sampling. The conclusion is not tied to a particular architecture or to any specific reduction primitive: it applies to any token-reduction operator whose effect can be summarized by a stable local response $H_P(k)$ and that is applied identically to all classes and timesteps so that the paired estimator remains valid.

# B IMPLEMENTATION DETAILS

## B.1 DATASETS AND EVALUATION PROTOCOLS

### B.1.1 DATASET DETAILS

We evaluate on four widely-used benchmarks, summarized in Table 9. Following Li et al. (2023), ImageNet-1K is sub-sampled to 2,000 images for classification to reduce computational cost, while the full validation set is retained for generation experiments.

Table 9: Dataset statistics with official splits used in our experiments.

| Dataset | Classes | Split | # Images (Cls.) | # Images (Gen.) |
|---|---|---|---|---|
| ImageNet-100(Tian et al., 2020) | 100 | Val. | 5,000 | 5,000 |
| ImageNet-1K (Russakovsky et al., 2015) | 1,000 | Val. | 2,000 | 50,000 |
| Oxford-IIIT Pets (Parkhi et al., 2012) | 37 | Test | 3,669 | 3,669 |
| COCO-2017 (Lin et al., 2014) | 80 | Val. | 5,000 | 5,000 |

### B.1.2 DIFFUSION CLASSIFIER PROTOCOL

**Diffusion-classifier.** We follow the *Diffusion Classifier* framework (Li et al., 2023), which scores a candidate conditioning $c$ by the expected noise-prediction error $\mathbb{E}_{t,\epsilon}\big[\|\epsilon - \epsilon_\theta(x_t, c)\|_2^2\big]$ and selects the minimizer. This method is *training-free*, requiring no calibration or finetuning, and enables

zero-shot classification directly from pretrained diffusion models. To enable evaluation on large label spaces, we use adaptive evaluation with staged pruning (detailed in Algorithm C.1). We adjust only `TrialList` and `KeepList` based on the size of the candidate set.

Table 10: Adaptive diffusion-classifier parameters per dataset. $N_{\text{stages}}$ is the number of pruning stages; `TrialList` is the cumulative number of Monte Carlo trials per candidate by stage; `KeepList` is the number of candidates retained after each stage.

| Dataset | $N_{\text{stages}}$ | TrialList | KeepList |
|---|---|---|---|
| ImageNet-100 | 2 | [5, 20] | [5, 1] |
| COCO-2017 | 2 | [5, 20] | [5, 1] |
| Oxford-IIIT Pets | 2 | [5, 20] | [5, 1] |
| ImageNet-1K | 3 | [5, 20, 100] | [50, 10, 1] |

For completeness, we also evaluated velocity-prediction flow-matching models (FLUX (Labs et al., 2025)). Using the `FlowMatchEulerDiscreteScheduler` to construct affine mappings for recovering $\hat{\epsilon}_\theta$ and $\hat{x}_0$ within DDIM, the released FLUX.1-dev checkpoint performed only marginally better than random guessing under the diffusion-classifier protocol. To avoid adapter-specific confounds and ensure a fair comparison, we restrict all evaluations to standard noise-prediction models.

## B.2   MODEL CONFIGURATIONS

### B.2.1   PROMPT TEMPLATES

For the classification task, following (Li et al., 2023), we use ``a photo of a {class}'' for ImageNet and COCO datasets, and ``a photo of a {class}, a type of pet'' for Oxford-IIIT Pets.

For generation, we use the same templates except for COCO-2017, where we use the official validation captions.

### B.2.2   GENERATION SETUP

We standardize generation across both backbones. For Stable Diffusion 2.0 (UNet) (Rombach et al., 2021), we use the EulerDiscreteScheduler with a scaled-linear beta schedule (beta_start 0.00085, beta_end 0.012, 1,000 training steps, epsilon prediction). For DiT-XL/2-512 (Peebles & Xie, 2023), we use the DDIMScheduler with a linear beta schedule (beta_start 0.0001, beta_end 0.02, 1,000 training steps, epsilon prediction). In both cases, we sample for 50 steps at 512×512 resolution. We apply classifier-free guidance with a scale of 7.5 for Stable Diffusion 2.0 and 4.0 for DiT-XL/2-512. Unless otherwise stated, all experiments are conducted in FP16 precision. For evaluation, FID scores are computed using the `pytorch-fid` implementation (Seitzer, 2020).

## B.3   TOKEN COMPRESSION

### B.3.1   COMPRESSION SETTINGS

Guided by the ablation in Table 6, we apply compression exclusively to self-attention (SA) and leave cross-attention (CA) and MLP blocks intact to preserve prompt adherence. For merging-based operators, merging is performed inside each SA block and an explicit unmerge restores the original sequence length before the residual addition, ensuring dense outputs for downstream modules. For KV-downsampling operators, only keys and values are subsampled while queries remain full-length, removing the need for unmerge.

**Stable Diffusion 2.0 (U-Net).** We insert compression exclusively at the highest-resolution encoder layers, where the spatial token count, and thus attention cost is maximal. This targets the primary bottleneck while maintaining quality.

**Diffusion Transformer (DiT-XL/2).** To assess generality beyond U-Net architectures, we port the same operators to DiT-XL/2. Specifically, token compression is applied within the first 12 transformer blocks, comparing early (blocks 1–6) versus mid-early (blocks 7–12) reduction, while leaving later blocks—where class conditioning and fine structural details consolidate unchanged.

### B.3.2 BASELINE IMPLEMENTATION

For all token compression baselines, we use the official implementations and default parameters released by the authors, and run them under a common experimental protocol (Sec. B.1.2, Sec. B.2.2) to ensure fair comparison and avoid unintentional re-tuning. The only modification we introduce is to vary the token reduction ratio, so that each method can be fairly evaluated under different levels of compression.

## B.4 EFFICIENCY EVALUATION

To measure the acceleration effect of our token reduction methods, we evaluate on the official Stable Diffusion 2.0 implementation released by Stability AI (Rombach et al., 2021). All experiments are conducted on a single NVIDIA RTX 4090 GPU in half-precision (`float16`). We report wall-clock sampling time per image batch excluding the VAE encoding/decoding overhead, since our methods target the denoising backbone rather than the autoencoder. FLOPs are measured using `FlopCounterMode` from `torch.utils.flop_counter` (Paszke et al., 2019). The corresponding runtime and efficiency results are summarized in Table 11.

Table 11: **Stable Diffusion 2.0 efficiency (batch size 4).** Wall-clock sampling time per *batch* (seconds) excluding VAE encode/decode. All rows use merge ratio $r = 0.7$.

| Method | Time ↓ (s / batch) | Acceleration ↑ (%) | FLOPs ↓ (G) |
|---|---|---|---|
| Baseline (No Accel.) | 11.98 | – | 804.26 |
| SiTo (Zhang et al., 2025) | 8.71 | 27.30 | 748.49 |
| ToMe (Smith et al., 2024) | 7.37 | 38.48 | 704.87 |
| Laplacian Gated Merge (Ours) | 7.37 | 38.48 | 704.99 |
| Cached Assignment Merge (Ours) | 7.29 | 39.15 | 698.88 |
| Adaptive Block Merging (Ours) | 7.27 | 39.32 | 695.08 |

# C ALGORITHM

## C.1 ADAPTIVE DIFFUSION CLASSIFIER

Naïve diffusion classification requires evaluating all candidate classes, and thus its cost grows linearly with the number of classes. To mitigate this, we adopt the adaptive evaluation strategy introduced in the diffusion-classifier framework (Li et al., 2023). At each stage, we allocate a fixed budget of trials across the remaining classes, discard unlikely candidates based on their average error, and retain only the most promising ones. This progressive pruning concentrates computation on high-confidence classes, enabling more fine-grained Monte Carlo error estimation. The procedure is summarized in Algorithm 1.

## C.2 FREQUENCY-AWARE TOKEN SCORING

*Spectral* structure of latent features is important for both discriminative and generative ability. High–frequency tokens encode the information of edges, textures, and small objects, especially at the late denoise stage, which are indispensable for recognition. However, high-frequency tokens can also amplify the variance in the diffusion classifier since predictions are aggregated over Monte Carlo draws of timesteps and noise; excess high–frequency tokens inflate the per–timestep estimation variance. Moreover, different timesteps emphasize different bands, early denoising focuses on low frequencies (global structure) while later steps emphasize high frequencies (fine detail). Therefore, the compression schedule should be *spectrally balanced and temporally consistent* to avoid injecting avoidable variance across timesteps. The necessity of preserving a balanced spectrum is confirmed empirically in Table 12, where discarding either high- or low-frequency tokens severely harms classification.

Our **BiGain**$_{TM}$ design follows from this principle. Since token merging resembles a local low–pass filter, we encourage merging only in small, spectrally smooth neighborhoods, where low–frequency information can be safely aggregated, while protecting detail-rich tokens that anchor class-critical

---

**Algorithm 1** Diffusion Classifier (Adaptive) (Li et al., 2023)

---

**Require:** test image $\mathbf{x}$, conditioning inputs $\mathcal{C} = \{\mathbf{c}_i\}_{i=1}^n$ (e.g., text embeddings or class indices), number of stages $N_{\text{stages}}$, list KeepList of number of $\mathbf{c}_i$ to keep after each stage, list TrialList of number of trials done by each stage

1: Initialize Errors$[\mathbf{c}_i]$ = list() for each $\mathbf{c}_i$
2: Initialize PrevTrials $= 0$     ▷ How many times we've tried each remaining element of $\mathcal{C}$ so far
3: **for** stage $i = 1, \ldots, N_{\text{stages}}$ **do**
4:     **for** trial $j = 1, \ldots,$ TrialList$[i]$ $-$ PrevTrials **do**
5:        Sample $t \sim [1, 1000]$
6:        Sample $\boldsymbol{\epsilon} \sim \mathcal{N}(0, I)$
7:        $\mathbf{x}_t = \sqrt{\bar{\alpha}_t}\mathbf{x} + \sqrt{1 - \bar{\alpha}_t}\boldsymbol{\epsilon}$
8:        **for** conditioning $\mathbf{c}_k \in \mathcal{C}$ **do**
9:           Errors$[\mathbf{c}_k]$.append($\|\boldsymbol{\epsilon} - \boldsymbol{\epsilon}_\theta(\mathbf{x}_t, \mathbf{c}_k)\|^2$)
10:        **end for**
11:     **end for**
12:     $\mathcal{C} \leftarrow \underset{\substack{\mathcal{S} \subset \mathcal{C}; \\ |\mathcal{S}| = \text{KeepList}[i]}}{\arg\min} \sum_{\mathbf{c}_k \in \mathcal{S}} \text{mean}(\text{Errors}[\mathbf{c}_k])$     ▷ Keep top KeepList$[i]$ conditionings
13:     PrevTrials $=$ TrialList$[i]$
14: **end for**
15: **return** $\underset{\mathbf{c}_i \in \mathcal{C}}{\arg\min} \text{mean}(\text{Errors}[\mathbf{c}_i])$

---

Table 12: **Classification results on frequency-based KV selection on ImageNet-100.** We compare the standard ToDo strategy with frequency-aware variants that select tokens with the highest or lowest Laplacian scores globally. Retaining only high- or low-frequency tokens severely degrades classification performance, highlighting the need to preserve a balanced spectrum.

| Downsampling strategy | Acc@1 ↑ | KV token sparsity |
|---|---|---|
| Todo (Nearest-Neighbor) (Smith et al., 2024) | 72.30 | 75% |
| Low-frequency tokens (lowest-laplacian) | 45.58 | 75% |
| High-frequency tokens (Highest-laplacian) | 26.56 | 75% |

microstructures. This balanced policy removes redundancy without sacrificing classification accuracy or generation fidelity. Practically, we introduce a set of fast, training-free scoring heuristics to decide which tokens to *preserve* (high detail) and which to *merge* (smooth/redundant), and we apply them consistently across timesteps so that each per-timestep classifier score remains reliable and contributes coherently to the Monte Carlo ensemble.

**Notation.** Let $\boldsymbol{X} \in \mathbb{R}^{H \times W \times C}$ denote the hidden feature tensor (height $H$, width $W$, channels $C$). For spatial index $(i, j)$, the token (channel vector) is $\boldsymbol{x}_{i,j} := \boldsymbol{X}_{i,j,:} \in \mathbb{R}^C$. The global mean token is $\boldsymbol{\mu} := \frac{1}{HW} \sum_{p=1}^H \sum_{q=1}^W \boldsymbol{x}_{p,q}$. For a 3×3 spatial kernel $\boldsymbol{L}$, $(\boldsymbol{X} * \boldsymbol{L})_{i,j,c}$ denotes 2-D convolution at $(i, j)$ on channel $c$. Let $\mathbb{N}_4(i, j)$ be the (in-bounds) 4-neighborhood of $(i, j)$ (up/down/left/right). The DFT of $\boldsymbol{x}_{i,j}$ at channel–frequency bin $k$ is $\hat{\boldsymbol{x}}_{i,j,k} := \sum_{c=1}^C (\boldsymbol{x}_{i,j})_c \, e^{-2\pi i (c-1)(k-1)/C}$ for $k \in \{1, \ldots, C\}$. We write $|| \cdot ||_p$ for the vector $\ell_p$ norm, $|| \cdot || \equiv || \cdot ||_2$, and $\langle \boldsymbol{a}, \boldsymbol{b} \rangle$ for the Euclidean inner product. We compute a scalar score $F_{i,j} \in \mathbb{R}$ per token, where larger values indicate detail-rich tokens and smaller values indicate smooth/redundant tokens. We list all functions of different metrics in Table 13.

For all heuristics except cosine-based ones, larger $F_{i,j}$ indicates stronger local variation and thus high-frequency detail. In contrast, for cosine similarity scores, *smaller* values correspond to tokens that deviate more from their neighbors or the global mean, and are therefore detail-rich.

Table 13: Formulas of different metrics.

| Metric Name | Formula |
|---|---|
| Global mean deviation | $F_{i,j} = \|\boldsymbol{x}_{i,j} - \boldsymbol{\mu}\|$ |
| $\ell_1$ norm | $F_{i,j} = \|\boldsymbol{x}_{i,j}\|_1$ |
| $\ell_2$ norm | $F_{i,j} = \|\boldsymbol{x}_{i,j}\|$ |
| Channel variance | $F_{i,j} = \frac{1}{C} \sum_{c=1}^{C} \left( (\boldsymbol{x}_{i,j})_c - \frac{1}{C} \sum_{c'=1}^{C} (\boldsymbol{x}_{i,j})_{c'} \right)^2$ |
| Laplacian ($\ell_1$) | $F_{i,j} = \frac{1}{C} \sum_{c=1}^{C} |(\boldsymbol{X} * \boldsymbol{L})_{i,j,c}|, \quad \boldsymbol{L} = \begin{bmatrix} 0 & 1 & 0 \\ 1 & -4 & 1 \\ 0 & 1 & 0 \end{bmatrix}$ |
| Laplacian ($\ell_2$) | $F_{i,j} = \sqrt{\frac{1}{C} \sum_{c=1}^{C} \left( (\boldsymbol{X} * \boldsymbol{L})_{i,j,c} \right)^2}$ |
| DFT spectral centroid | $F_{i,j} = \frac{\sum_{k=1}^{C} k\, |\hat{\boldsymbol{x}}_{i,j,k}|}{\sum_{k=1}^{C} |\hat{\boldsymbol{x}}_{i,j,k}|}$ |
| DFT total amplitude | $F_{i,j} = \sum_{k=1}^{C} |\hat{\boldsymbol{x}}_{i,j,k}|$ |
| Cosine similarity to neighbors | $F_{i,j} = \frac{1}{|\mathbb{N}_4(i,j)|} \sum_{(p,q) \in \mathbb{N}_4(i,j)} \frac{\langle \boldsymbol{x}_{i,j}, \boldsymbol{x}_{p,q} \rangle}{\|\boldsymbol{x}_{i,j}\| \|\boldsymbol{x}_{p,q}\|}$ |
| Cosine similarity to global mean | $F_{i,j} = \frac{\langle \boldsymbol{x}_{i,j}, \boldsymbol{\mu} \rangle}{\|\boldsymbol{x}_{i,j}\| \|\boldsymbol{\mu}\|}$ |

## C.3 BIGAIN$_{\text{TM}}$

Algorithm 2 presents our frequency-aware token merging method. The core innovation lies in using spectral information to guide merge decisions, ensuring that token reduction preserves both generative fidelity and discriminative utility. The algorithm first applies a frequency scorer $\mathcal{F}$ (default: Laplacian filtering C.2) to identify local frequency content in the spatial feature map. Tokens with low frequency scores indicate smooth, homogeneous regions amenable to merging, while high scores correspond to edges, textures, and fine details critical for classification.

The destination selection step partitions the spatial layout into regular grids and identifies the lowest-frequency token within each grid as a merge destination. This strategy ensures spatial coverage while directing merging toward spectrally smooth regions. The remaining tokens form a source set, which is then assigned to destinations via bipartite matching based on cosine similarity. By selecting the top-$r$ fraction of most similar pairs, the method preserves semantic coherence while respecting the frequency-based partitioning. After merging and processing through attention layers, an unmerge operation restores the original sequence length for architectural compatibility.

---

**Algorithm 2 BiGain$_{\text{TM}}$**: Frequency-Aware Token Merging

---

**Require:** Tokens $\boldsymbol{X} \in \mathbb{R}^{N \times d}$, merge ratio $r$, grid size $s$, frequency scorer $\mathcal{F}$
1: **function** BIGAINMERGE($\boldsymbol{X}, r, s, \mathcal{F}$)
2: $\quad \boldsymbol{f} \leftarrow \mathcal{F}(\boldsymbol{X})$ $\qquad\qquad\qquad\qquad\qquad\qquad$ ▷ Score tokens by frequency content
3: $\quad \mathbb{D} \leftarrow \text{SelectDestinations}(\boldsymbol{f}, s)$ $\qquad\qquad\qquad\qquad$ ▷ Lowest frequency per grid
4: $\quad \mathbb{S} \leftarrow \{1, \ldots, N\} \setminus \mathbb{D}$ $\qquad\qquad\qquad\qquad$ ▷ Remaining tokens as sources
5: $\quad \mathcal{M} \leftarrow \text{BipartiteMatch}(\boldsymbol{X}_{\mathbb{S}}, \boldsymbol{X}_{\mathbb{D}}, r)$ $\qquad\qquad$ ▷ Similarity-based assignment
6: $\quad \boldsymbol{X}^{\text{merged}} \leftarrow \text{Merge}(\boldsymbol{X}, \mathcal{M})$ $\qquad\qquad\qquad$ ▷ Combine assigned tokens
7: $\quad \boldsymbol{Z} \leftarrow \text{Process}(\boldsymbol{X}^{\text{merged}})$ $\qquad\qquad\qquad\qquad\qquad$ ▷ Apply attention
8: $\quad$ **return** Unmerge($\boldsymbol{Z}, \mathcal{M}$) $\qquad\qquad\qquad\qquad$ ▷ Restore dimensions
9: **end function**

---

Algorithm 3 presents Adaptive Block Merge (ABM), a computationally efficient variant designed for high-resolution stages where token count is maximal. Rather than per-token assignment, ABM operates at block granularity. After computing frequency scores, the feature map is partitioned into blocks, and blocks are ranked by their frequency content. The lowest-scoring fraction $r$ of blocks are identified as smooth regions and merged via averaging, while high-frequency blocks remain intact. This block-level decision reduces computational complexity of bipartite matching, providing speedup with little accuracy degradation as demonstrated in our Table 8.

**Algorithm 3** Adaptive Block Merge (ABM): Fast **BiGain$_{\text{TM}}$** Variant

---

**Require:** Tokens $\boldsymbol{X} \in \mathbb{R}^{N \times d}$, block size $b$, merge ratio $r \in [0, 1]$, scorer $\mathcal{F}$
1: **function** ADAPTIVEBLOCKMERGE($\boldsymbol{X}, b, r, \mathcal{F}$)
2:     $\boldsymbol{f} \leftarrow \mathcal{F}(\boldsymbol{X})$                                     ▷ Compute frequency scores
3:     $\mathcal{B} \leftarrow$ BlockPartition($\boldsymbol{X}, b$)                    ▷ Partition into $b \times b$ blocks
4:     $\mathcal{B}_{\text{smooth}} \leftarrow$ SelectLowestFreq($\mathcal{B}, \boldsymbol{f}, r$)         ▷ Select lowest $r$ fraction blocks
5:     $\boldsymbol{X}^{\text{merged}} \leftarrow$ MergeBlocks($\boldsymbol{X}, \mathcal{B}_{\text{smooth}}$)              ▷ Average selected blocks
6:     $\boldsymbol{Z} \leftarrow$ Process($\boldsymbol{X}^{\text{merged}}$)                         ▷ Apply attention
7:     **return** RestoreBlocks($\boldsymbol{Z}, \mathcal{B}_{\text{smooth}}$)                  ▷ Restore dimensions
8: **end function**

---

## C.4   BiGain$_{\text{TD}}$

Algorithm 4 presents our Interpolate–Extrapolate KV-Downsampling method, which reduces attention complexity by downsampling keys and values while preserving queries at full resolution. This asymmetric approach maintains the model's ability to attend precisely to all spatial positions while reducing memory and computation. The key innovation is the controllable linear combination of nearest-neighbor and average pooling, allowing fine-grained control over the frequency-preservation trade-off.

Here we use the same interpolate–extrapolate operator $D_{\alpha,s}$ as defined in Eq. 8. This operator blends nearest-neighbor sampling (preserving detail) with average pooling (smoothing), controlled by the parameter $\alpha \in \mathbb{R}$. Keys and values are downsampled as $\tilde{K} = D_{\alpha,s}(K)$ and $\tilde{V} = D_{\alpha,s}(V)$, while queries remain full resolution.

**Algorithm 4 BiGain$_{\text{TD}}$**: Interpolate–Extrapolate KV-Downsampling (IE-KVD)

---

**Require:** Tokens $\boldsymbol{X} \in \mathbb{R}^{N \times d}$, downsample factor $s$, interpolation-extrapolation $\alpha \in \mathbb{R}$, scorer $\mathcal{F}$
1: **function** BIGAINDOWNSAMPLE($\boldsymbol{X}, s, \alpha, \mathcal{F}$)
2:     $\boldsymbol{f} \leftarrow \mathcal{F}(\boldsymbol{X})$                                     ▷ Compute frequency scores
3:     $\boldsymbol{Q} \leftarrow \boldsymbol{X}\boldsymbol{W}_Q$                                  ▷ Compute queries (full resolution)
4:     $\boldsymbol{K} \leftarrow \boldsymbol{X}\boldsymbol{W}_K, \boldsymbol{V} \leftarrow \boldsymbol{X}\boldsymbol{W}_V$              ▷ Compute keys and values
5:     $\tilde{K} \leftarrow$ Interpolate/ExtrapolateDownsample($\boldsymbol{K}, s, \alpha, \boldsymbol{f}$)         ▷ Downsample K
6:     $\tilde{V} \leftarrow$ Interpolate/ExtrapolateDownsample($\boldsymbol{V}, s, \alpha, \boldsymbol{f}$)         ▷ Downsample V
7:     $\boldsymbol{Z} \leftarrow$ Attention($\boldsymbol{Q}, \tilde{K}, \tilde{V}$)              ▷ Q at full res, K/V downsampled
8:     **return** $\boldsymbol{Z}$                                     ▷ Output maintains full resolution
9: **end function**

---

# D   USE OF LARGE LANGUAGE MODELS

We used an LLM to help solely polish the writing of the paper, while all ideas and experiments are conceived and carried out entirely by the authors.

