# OpenReview forum: "BiGain: Unified Token Compression for Joint Generation and Classification"
_ICLR.cc/2026/Conference — ICLR 2026 Conference Withdrawn Submission_

### Official Review · Reviewer_Bo1j · 2025-10-24

**Soundness:** 2
**Presentation:** 2
**Contribution:** 2
**Rating:** 4
**Confidence:** 4

**Summary:**

This paper proposed BiGain, a token compression framework that enhances both generation and classification performance of diffusion models. They proposed Laplacian-gated token merging for merging among spectrally smooth tokens while discouraging merges of high-contrast tokens. They also proposed Interpolate-Extrapolate KV Downsampling for downsampling compression technique. Experiments on SD2.0 and DiT across different datasets show that BiGain achieves better performance on both generation FID and classification accuracy.

**Strengths:**

1.The method proposed in the paper has a certain performance improvement compared to baseline methods on different datasets.

2.The implementation of the paper method is not complicated and can be easily adapted to different models.

3.The paper conducted experiments on multiple datasets.

**Weaknesses:**

1.The motivation of the paper is not clear enough, why should the diffusion model be used for a large number of discriminative tasks instead of focusing on generative tasks. And motivation for separating features into frequency domain is not obvious.

2.The description of the method in the paper is unclear and lacks necessary formal language and symbolic definitions.

3.The paper lacks necessary visualization and illustrations to illustrate the motivation and implementation process of the proposed method.

4.The paper conducted a large number of experiments on discriminant analysis, especially classification, but the experiments on the generation part were too rudimentary. Evaluating the generation performance solely based on FID is insufficient, the paper should report more metrics like sFID, Precision, Recall and CLIP-Score.

5.The paper should conduct experiments on larger scale DiT architectures such as PixArt and Flux to demonstrate its effectiveness on more advanced models.

**Questions:**

Please see above weaknesses.

---

### Official Review · Reviewer_KvvT · 2025-10-30

**Soundness:** 3
**Presentation:** 2
**Contribution:** 3
**Rating:** 4
**Confidence:** 2

**Summary:**

This paper redefines the objective for token reduction in diffusion models, proposing a time-step local, frequency-based method that achieves improved performance on datasets.

**Strengths:**

1. Innovative Problem Definition: The paper's primary strength lies in redefining the goal of token reduction for diffusion models, shifting from a single focus on generation quality to a dual objective that also includes discriminative performance. This is a novel and important perspective.
2. Strong Empirical Results: The experiments are comprehensive and demonstrate that the proposed method achieves significant improvements over baselines on multiple datasets and model architectures.

**Weaknesses:**

1. Insufficient Analysis of the Core Conflict: Diffusion models can be used as training-free classifiers, which implies a high correlation between their generative and discriminative capabilities. The paper fails to provide a clear analysis of why previous methods, which primarily target generative ability, cause such a severe degradation in discriminative performance. This foundational analysis is missing.
2. Limited Novelty in Dual-Objective Design: The only design in the proposed method that specifically addresses the dual objective is its time-step local nature. Fulfilling this characteristic is neither particularly difficult nor original. Consequently, the design considerations specifically for the discriminative objective appear insufficient and underdeveloped.
3. Poor Organization.There are some layout issues in the PDF.

**Questions:**

1. Please explain the main difficulties encountered after introducing the dual objective. How does your method solve them (i.e., what is your specific contribution in this regard)?
2. What is the relationship between the frequency-based method and the dual-objective problem it aims to solve?

---

### Official Review · Reviewer_3gUP · 2025-10-31

**Soundness:** 3
**Presentation:** 2
**Contribution:** 3
**Rating:** 4
**Confidence:** 4

**Summary:**

This paper introduces BiGain, a training-free, plug-and-play framework for accelerating diffusion models that jointly optimizes both generative quality and discriminative (classification) performance. Unlike previous acceleration methods that focus only on synthesis quality, BiGain uses frequency-aware token compression to preserve both fine details and global semantics. It proposes two novel operators: Laplacian-gated token merging (which merges spectrally smooth tokens, preserving edges/textures) and Interpolate–Extrapolate KV Downsampling (which downsamples attention keys/values while maintaining query precision). BiGain consistently improves the speed–accuracy trade-off for diffusion-based classification and generation across various datasets and architectures, demonstrating significant gains in both tasks without retraining.

**Strengths:**

The paper proposes a simple yet effective framework, BiGain, for unified token compression in diffusion models, addressing both generation and classification. The experimental design is thorough and well-executed—extensive experiments across multiple architectures (U-Net, DiT) and datasets (ImageNet, COCO, Oxford-IIIT Pets) convincingly demonstrate the method’s effectiveness. The analysis is comprehensive, with ablation studies and visualizations that provide clear insights into the model’s behavior. Furthermore, the appendix presents well-formulated mathematical derivations and theoretical proofs, offering a rigorous foundation that helps readers better understand the algorithm’s principles.

**Weaknesses:**

While the paper presents extensive quantitative comparisons (e.g., FID, accuracy), the evaluation primarily focuses on numerical metrics. For a generation-related work, qualitative assessment is equally crucial—especially when employing lossy acceleration strategies such as token compression. However, the paper lacks visual examples of generated results, making it difficult to judge the actual perceptual quality and aesthetic fidelity of the outputs. In many cases, FID may remain stable while the visual quality deteriorates noticeably.

Additionally, the paper would benefit from a clear and illustrative teaser figure to concisely convey the overall idea and workflow of the proposed method. This would greatly enhance the paper’s readability and help readers quickly grasp the core contributions.

**Questions:**

In Figure 1, it is interesting to observe that when the merge ratio is small or the downsampling factor is set to 2, the accelerated models slightly outperform the uncompressed baseline. Could the authors clarify this phenomenon? Is it merely due to random variation, or does it suggest a genuine benefit from mild token compression—such as implicit regularization or feature denoising that improves discriminative or generative performance?

---

### Official Review · Reviewer_Vjmt · 2025-11-01

**Soundness:** 2
**Presentation:** 1
**Contribution:** 2
**Rating:** 2
**Confidence:** 3

**Summary:**

The paper presents BiGain, a training-free, spectrum-aware token compression framework for diffusion models (DMs) that enables efficient deployment without compromising generative fidelity or discriminative accuracy. The framework introduces two key components: L-GTM and IE-KVD, performing per-timestep adaptive compression while maintaining compatibility with diffusion classifiers’ paired-sampling estimators. Extensive experiments demonstrate that BiGain achieves significant speedups and memory savings while maintaining—or even improving—FID/IS and stabilizing classification accuracy under heavy compression. The method is plug-and-play and requires no fine-tuning, providing a practical route toward faster, lighter diffusion inference.

**Strengths:**

1. The paper reframes token compression for diffusion models from a generation-only objective to a joint generation and discrimination objective.
2. From a frequency-domain perspective, it introduces the principle of balanced spectral retention and clearly explains why common acceleration methods tend to degrade classification accuracy earlier and more severely.
3. The empirical coverage is extensive, with comprehensive comparisons and ablations that substantiate the core claims.

**Weaknesses:**

1. The manuscript assumes a scenario where diffusion models must perform both generation and classification, proposing a unified lightweight compression framework for such dual-purpose usage. However, this setting appears questionable, as in practice generation and classification are typically deployed as separate models with distinct optimization objectives and performance requirements. The motivation for joint compression is therefore unclear and may not correspond to a realistic deployment setting.
2. The claims are weakly justified and loosely connected to the method’s motivation. For instance, in line 76, the paper asserts that diffusion classifiers focus on low-frequency components at early timesteps and on high-frequency components at later ones—an unsubstantiated claim. The subsequent motivation for cross-timestep integration lacks a logical transition from this frequency-based argument, and such issues do not seem to arise in DiT-like architectures. Furthermore, the argument for maintaining temporal consistency to reduce Monte Carlo variance is again introduced without clear conceptual linkage.
3. The presentation lacks clarity. The manuscript would benefit from illustrative diagrams that visually explain the workflow and how each module operates within the framework.

**Questions:**

1. Why is it necessary for a single diffusion model to handle both generation and classification tasks, rather than deploying separate lightweight models for each? Could the authors provide a concrete example of a real-world application that requires this joint setting?
2. How can a jointly compressed model outperform methods optimized for a single task (generation or classification) when evaluated on that specific task?
3. In Table 1, the TD variant consistently outperforms the TM variant. What is the practical or conceptual motivation for retaining the TM version?

---

### Note · Authors · 2025-11-12

I have read and agree with the venue's withdrawal policy on behalf of myself and my co-authors.